# Enhancement of Skin Wound Healing by rhEGF-Loaded Carboxymethyl Chitosan Nanoparticles

**DOI:** 10.3390/polym12071612

**Published:** 2020-07-20

**Authors:** Pei Zhang, Chenguang Liu

**Affiliations:** 1College of Marine Life Sciences, Ocean University of China, Qingdao 266003, China; zhangpei8877@126.com; 2Department of Life Science, Luoyang Normal University, Luoyang 471934, China

**Keywords:** carboxymethyl chitosan, conjugated linoleic acid, self-assembly, nanoparticles, cytotoxicity, rhEGF, migration, re-epithelialization

## Abstract

The self-assembly of hydrophobically modified polymers has become a research hotspot due to its wide application in the biomedical field. Recombinant human epidermal growth factors (rhEGFs) are molecules that are able to enhance wound healing; however, they have a short half-life and require sustained action to enhance their mitogenic effect on epithelial cells. Here, we proposed a new delivery system to avoid the inhibition of rhEGF by various enzymes, thus improving its bioavailability and sustained release. The amphiphilic polymer was composed of conjugated linoleic acid (CLA) and carboxymethyl chitosan (CMCS), which were further characterized by fourier transformed infrared spectroscopy (FTIR) and ^1^H nuclear magnetic resonance (^1^H NMR). Then, the self-assembly behavior of CLA–CMCS (CC) polymer in water was observed in which the particle size of CC decreased from 196 to 155 nm with the degree of CLA substitution increasing. The nanoparticles were loaded with rhEGF and the maximum rhEGF loading efficiency (LE) of CC3 nanoparticles was 82.43 ± 3.14%. Furthermore, CC nanoparticles (NPs) exhibited no cytotoxicity for L929 cells, and cell proliferation activity was well preserved after rhEGF loading to CC-NPs and was comparable to that of free rhEGF. Topically applied rhEGF:CC-NPs significantly accelerated the wound-closure rate in full thickness, which was most probably due to its sustained release and enhanced skin permeation. In conclusion, carboxymethyl chitosan-based nanoparticles were constructed and showed good cytocompatibility. Moreover, these findings also demonstrated the therapeutic potential of rhEGF:CC-NPs as a topical wound-healing drug carrier.

## 1. Introduction

Skin wound healing is a complex and long process involving coagulation, inflammation, and the formation of new skin tissue [1]. In particular, burn wound healing and chronic wound healing remain some of the challenging and costly health-care issues around the world [2,3]. In order to solve this problem, many growth factors (GF), such as platelet-derived growth factor (PDGF), epidermal growth factor (EGF), transforming growth factor family (TGF), and fibroblast growth factor (FGF), have been identified as promoters of skin wound healing and play important roles in successful tissue repair [4,5,6,7]. The supply of exogenous growth factors may lead to faster re-epithelialization, resulting in decreased infection risk [8]. The use of growth factors in chronic wound treatment can promote wound closure and restore abnormalities in the healing process. For example, EGF is an important kind of GF that can promote the proliferation of keratinocytes in vivo and accelerate wound healing [9]. However, the therapeutic effect of EGF is greatly restricted due to its short half-life, toxicity in high concentration, and physical and chemical instability in the body when EGF is administrated via injection or in free form [10]. Furthermore, a lot of proteases are activated and easily decompose EGF at the wounded site [11]. Therefore, it should be mentioned that devising an efficient delivery method plays a vital role in the application of EGF.

Bioderived materials-based nanoparticles have gained great interest and been developed into effective drug delivery systems with good biocompatibility, nontoxic catabolites, and microinflammation characteristics [12]. Since lipid nanoparticles were first reported as drug delivery systems in the 1990s, they have attracted wide attention as effective and nontoxic carriers of various drug molecules [13]. Self-assembled nanoparticles based on poly(lactic–*co*–glycolic acid) (PLGA) and polysaccharides, which are composed of an inner hydrophobic core and outer hydrophilic shell, have been widely reported to improve drug delivery efficiency [14,15,16]. These nanoparticles are also suitable delivery systems for skin wound treatment due to their high drug concentration in the treatment area and the ability to induce tissue repair. In addition, their nano size ensures that the particles are in close contact with the skin, while controlling the release of the drug to increase the action time of the drug on the skin surface [17]. These particles also show occlusive properties, increasing skin hydration and enhancing drug permeability [18].

Here, we linked hydrophobic conjugated linoleic acid (CLA) to carboxymethyl chitosan (CMCS) molecules with the intention to form amphiphilic CLA–CMCS (CC), which can be used as a protein-based nanocage. The formation of CC-NPs was investigated, and their physicochemical properties were characterized, such as size, polydispersity (PDI), and zeta potential. The cytotoxicity of CC-NPs (nanoparticles) was also investigated to select the optimal concentration of drug loading system. Recombinant human epidermal growth factors (rhEGF) was loaded into CC-NPs as a model protein, and the bioactivity of the loaded rhEGF was tested with fibroblasts culture assays (compared with free rhEGF). Furthermore, the rhEGF-loaded CC-NPs were applied to the full thickness excised wound model of mice, and the wound healing was evaluated according to wound closure, inflammatory recovery, and epithelial regeneration.

## 2. Materials and Methods

### 2.1. Materials and Cell Line

CMCS (substitution degree of carboxymethyl >80%, viscosity = 600 mpa·s) was obtained from Sigma Co. Ltd. (St. Louis, MO, USA). 1-Ethyl-3-(3-dimethylaminopropyl) carbodiimide (EDC), CLA, dulbecco’s modified eagle medium (DMEM) medium were from Solarbio Co. Ltd. (Beijing, China). Cell Culture Flasks (25 cm) and cell culture plates were from Costar Co. Ltd. (New York, NY, USA).

A L929 cell line purchased from Shanghai cell bank was used in the experiment. L929 cells were cultured in DMEM supplemented with 10% heated-inactivated fetal bovine serum (FBS), 100 U/mL penicillin, 100 μg/mL streptomycin, and 2 mM glutamine. The culture environment of the cells was that they were incubated at 37 °C in a humid atmosphere of 5% CO_2_ in 95% air.

### 2.2. Synthesis of CLA–CMCS Conjugates

As mentioned in our previous study, CLA can be coupled to CMCS by forming the new amide bond [19]. Firstly, CMCS was dissolved in aqueous solution at a final concentration of 1% (*w/v*). CLA was dissolved in 15 mL methanol and then added to the CMCS solution drop wise while stirring at room temperature. After 24 h, the reactants were dialyzed with dialysis bags to remove impurities and unreacted substances and then freeze-dried for 24 h. CC products with different CLA substitution degrees were named CC1, CC2, and CC3, respectively.

The CC conjugates were characterized by fourier transformed infrared spectroscopy (FTIR) and ^1^H nuclear magnetic resonance (^1^H NMR). The IR spectra of lyophilized CC conjugates were measured on a Nexus 470 FTIR spectrometer (Nicolet, MN, USA) at 25 °C. Powder samples were blended with high quality and dry KBr, and then pressed into disks for measurement. The ^1^H NMR spectrum of the conjugates was generated by a 500-MHz NMR (UNIYTINOVA-500 NMR, VARIAN) at 25 °C. A CC sample was dissolved in solution of NaOD (analytical reagent, Sigma), yielding a concentration of 10 mg/mL. The degree of substitution (DS) of CC conjugates were determined by elemental analysis (C, N) using a CE-440 elemental analysis instrument.

### 2.3. Preparation and Characterization of CC-NPs

The CC conjugates were repeatedly dissolved in distilled water and stirred for 24 h. Then, the solution was sonicated by a probe-type sonifier (Sonics Ultrasonic Processor, VC750, Hartford, CT, USA) at 200 W for 10 min each time, in which the pulse was turned off for 2.0 s with the interval of 2.0 s to prevent excessive temperature.

The morphology of CC-NPs was observed by transmission electron microscopy (TEM, H-600a, Hitachi, Japan). The sample was diluted with distilled water and placed on a copper mesh, which was air dried and negatively stained with 1% (*w*/*v*) phosphotungstic acid before observation. The CC-NPs were dispersed in 0.2 M phosphate-buffered solution (PBS, pH 7.4) and sonicated to determine the particle size, polydispersity (PDI), and zeta potential using a Malvern Zetasizer (3000HSA, Malvern, UK) at 25 °C.

CC-NPs solution (3 mL) was taken to determine the absorbance (OD) of the nanoparticle solution at a wavelength of 600 nm by spectrophotometry and use PBS buffer (pH 7.4) as a control to detect its light transmittance.

Hemolysis assay was performed using rabbit blood. A 0.5 mL aliquot of the erythrocyte stock dispersion was added per milliliter of CC-NP samples. At predetermined intervals, the oxyhemoglobin level was measured spectrophotometrically at 540 nm.

The in vitro occlusion test was adapted from Vringer’s protocol and repeated three times (n = 3). The occlusion factor (OF) is calculated by Formula (1):*OF* = (*W*_a_ − *W*_b_)/*W*_a_ × 100,(1)
where *W*_a_ is the water loss without sample (reference), and *W*_b_ is the water loss with sample.

### 2.4. In Vitro Studies

#### 2.4.1. Drug Entrapment Efficiency and Release Studies

After centrifugation, the content of rhEGF was determined at 275 nm by a uvikon 940 spectrophotometer (Kontron instruments, Eching, Germany). The amount of remaining drugs is detected in the supernatant, and the amount of loaded drugs was determined by subtracting the amount of remaining free drugs from the total dosage. The loading efficiency (LE) could be calculated by the following Equation (2):*LE* (%) = (*W*_total drug_ − *W*_free drug_/*W*_total drug_) × 100,(2)
where “*W*_total_
_drug_” is the total amount of all drugs, and “*W*_free drug_” is the amount of remaining free drugs detected in the supernatant after centrifugation. 

A Franz diffusion cell with a diameter of 0.9 cm (crown scientific, Sommerville, MA, USA) was used for release study. A 0.1 μm nitrocellulose membrane (sartorius, Gottingen, Germany) was coated on the Franz diffusion cell. Acetate buffer (pH 6.0) was used as receptor solution and CC-NPs dispersing solution was used as donor solution. The samples were collected at different time points in a day (250 μL each time) and finally detected at 275 nm using the spectrophotometer.

#### 2.4.2. Cell Compatibility Assay

The cell cytotoxicity of CC-NPs was determined using a 3-(4,5-dimethyl-2-thiazolyl)-2,5-diphenyl-2-H-tetrazolium bromide (MTT) assay [12]. L929 cells at 1 × 10^5^ cells/mL were seeded onto 96-well plates and allowed to adhere. The culture medium was replaced with a new one containing blank CC-NPs, in which the concentration range of NPs were 50–1000 μg/mL. Cells were further incubated for 24 h and then, MTT solution was added to each well. After 4 h of incubation, the medium was removed, and any crystal was dissolved in DMSO. A microplate reader was used to measure the absorbance of each hole at 490 nm after shaking slowly for 5 min. Cell survival was expressed as a percentage of the control (only test cells added).

#### 2.4.3. Migration Assay

L929 cells at 1 × 10^5^ cells/mL were seeded onto 96-well plates in DMEM and cultured at 37 °C for 24 h to form a confluent monolayer. Gaps approximately 0.9 mm in width were created by scratching the plates with a sterile pipette tip. L929 cells were seeded on 96-well plates at a concentration of 1 × 10^5^ cells/mL and cultured at 37 °C for 24 h to form a cell monolayer. Then, the plate was scraped with the suction head of the sterile pipette to form a 0.9 mm wide gap [20]. The medium was immediately removed and replaced with 80 μg/L of free rhEGF in serum-free DMEM, 40 or 80 μg/L rhEGF obtained from 1 day in vitro release studies (rhEGF: CC-NPs) in serum-free supplemented DMEM, and fresh serum-free medium. The average width of the gap is calculated using Image J software from the images taken by the microscope at three different positions of each hole. Cell migration pictures were taken at 0, 6, and 12 h after scratching. Furthermore, cell morphology was recorded during 24 h by microscope (Biostation IMQ, Nikon, Japan) in order to observe cell migration and proliferation.

### 2.5. In Vivo Studies

#### 2.5.1. Animals Wound Induction

All animal experiments had been approved by the Medicine Inspecting Institute of Qingdao, China. The rats were kept in separate cages for 12 h with light–dark cycles, and they were free to eat standard rodent food and water. A wound with a diameter of 2.54 cm^2^ was formed on the rats’ racing back.

#### 2.5.2. Serial Wound Analysis

After the wound was induced, a control group (PBS, pH = 7.4), empty CC-NPs group, free rhEGF group (80 μg/mL), and rhEGF:CC-NPs group (80 μg/mL) were topically applied to the wound area twice per day for 14 days. The body weight of the rats was measured three days before wounding and at days 0, 7, and 14 after wounding. The area of skin wound (cm^2^) was measured on the day of operation and the first, third, seventh, and 14th days after operation to determine the wound closure. When the wound area was equal to zero, the wound was considered to be completely closed [21].

#### 2.5.3. Histological Analysis

After euthanasia, the skin of rats was removed and fixed with 4% paraformaldehyde. After embedding with paraffin, 4.5 mm thick tissue sections were examined by optical microscope. Hematoxylin and eosin (H&E) staining was used to study the pathological changes of skin tissue [22].

### 2.6. Statistical Analysis

All the data in this study were expressed as mean ± SD. Data were analyzed statistically by the SPSS 24.0 programs software package. The chemical structure diagram of the polymer was drawn using ChemBioDraw Ultra 14.0 software, and * *p* < 0.05 was regarded as significant, ** *p* < 0.01 was considered as very significant.

## 3. Results and Discussion

### 3.1. Characterization of CC Sample

By changing the feed ratio of CLA to CMCS, various CC conjugates with different amounts of CLA were synthesized. CMCS and CC was analyzed using an FTIR spectrophotometer for the characteristic absorption bands, which are shown in Figure 1. The peaks at 1591, 1414 and 1070 cm^−1^ are all characteristic peaks of CMCS. They are assigned to the stretching vibrations of the C=O, CH_2_COO–, and –C–O– groups, separately [23], and the broad peak at 3339 cm^−1^ is for the –OH groups present both in CMCS and CC conjugates. In addition, the absorption peak at 2928 cm^−1^ represents the stretching vibration of C–H [20]. What’s more, the enhancement peak at 1736 cm^−1^ corresponding to the tensile vibration of the C=O group indicated the existence of a new ester bond, which further proves the successful preparation of CC conjugates [24,25].

As shown in Figure 2, the ^1^H NMR spectrum of CC3 further confirmed the formation of the new amide bond. The presence of the signal of a chitosan backbone at 1.0, 1.9, 2.9, and 3.5–3.8 ppm are assigned to the acetyl group, carbon 2, carbon 3-6, and carbon 1 of chitosan, respectively [26]. The peaks at 0.7–1.5, 2.3, 2.7, 4.6, 5.0, and 5.5 ppm were attributed to the pendant groups of CLA. In brief, the results of FTIR and ^1^H NMR showed that the CC conjugate was successfully synthesized.

### 3.2. Formation and Characteristics of CC-NPs

#### 3.2.1. Characterization of CC-NPs

The degree of substitution (DS) of CLA increased as the feed ratio of CLA increased, which was in the range from 3.7 to 6.5 (Table 1) per CMCS molecule in this experiment. The size and distribution of self-assembled nanoparticles were measured by dynamic laser scattering. The sizes of the majority of nanoparticles are 155–196 nm, which is significantly affected by the DS of CC conjugates. It suggests that the interactions between self-aggregates and the hydrophobic group ratio of CC conjugates could not be ignored [27]. The large zeta potential of CC self-assembly nanoparticles (about −20 mV) implies that the electrostatic repulsive force contributes a lot to the stability of the nanoparticles [28]. The CC3 sample with the highest DS showed the strongest transmittance compared with the other samples. The higher light transmittance indicated that the size of the nanoparticles was smaller and the degree of uniformity was higher, and there was no phenomenon of aggregation in the solution that could be observed by TEM. In addition, the *in vitro* hemolysis assay verified the good biocompatibility of CC-NPs, with CC3 being the most biocompatible sample. When the hemolysis rate of a material is less than 5%, it indicates that the material is suitable for use as a substrate for a drug carrier [29].

#### 3.2.2. TEM, Stability, and Occlusion Test of CC-NPs

TEM was used to evaluate the morphology of CC-NPs, and it showed that the self-assembled nanoparticles had spherical morphology and good structural integrity without any aggregation [30]. Its mean diameter was about 142 ± 3.83 nm, as shown in Figure 3A.

The stability of CC nanoparticles may be affected by environmental factors such as pH and temperature during their manufacture, storage, and transportation [31]. Therefore, the changes of particle size and size distribution were assessed at various pH and temperature. As can be seen from Figure 3B,C, the size of CC-NPs remained in the nanometer range in different pH and temperature environments. The particle size increased when the pH changed from 1.0 to 6.0, it reached the highest point at pH 6.0, and it decreased as the pH increased from 6.0 to 7.4 (Figure 3B). This suggests that CC nanoparticles possess the property of amphoteric electrolyte. Its isoelectric point (pI) is about 6.0. The large zeta potential of CC nanoparticles suggests that the electrical charge plays an important role in the stability of CC nanoparticles. At pH values that are close to pI, the electrostatic interaction became weaker, leading to flocculation and/or coalescence, and consequently an increase of droplet size. For all CC nanoparticles, the particle size decreased when the temperature rose from 4 to 32 °C (Figure 3C). These results agree with the results obtained by E.B. Souto on lipid nanoparticles [32]. PDI showed the same changing pattern as particle size in pH and temperature tests.

CC-NPs were utilized for an occlusion test, and the occlusion effect was measured at 6, 24, and 48 h (Figure 3D). With time extended, the occlusion factor increased in tests for all CC nanoparticles with different DS. This trend suggests that the occlusion effect became more obvious with time prolonged. This is in accordance with the occlusion test results obtained by other researchers [32]. CC3 is the most occlusive and CC2 is the least occlusive among these nanoparticles, which is observed at every time point in occlusion tests. The occlusive properties of CC3 may lead to its accumulation in the *Stratum*
*Corneum*, as well as the consequently sustained release of rhEGF and enhanced therapeutic effect [7].

### 3.3. In Vitro Studies

#### 3.3.1. rhEGF Loading Efficiency and Release Study

As can be seen in Table 1, the LE of CC-NPs was gradually increased from 72.16 ± 2.69% to 82.43 ± 3.14% with the increasing degree of hydrophobic substitutions in the CC conjugates. The increase of hydrophobic groups means that the hydrophobic microdomains in the self-assembled nanoparticles increase and have a stronger hydrophobic force, both of which can increase the encapsulation of epidermal growth factor by CC nanoparticles [33].

Figure 4 shows the release profile of rhEGF from rhEGF:CC-NPs at PBS7.4. It exhibited an initial burst of (68.84 ± 1.53)% in 2 h and an accumulative release of (85.50 ± 3.08)% after 24 h. The initial rapid release may be due to the dissolution of the surrounding rhEGF. The release profile in 10 h was observed, which was probably due to the full dissolution of the CMCS outer layer in the designated condition according to its polyampholytic character nature, and the internal embedded rhEGF is gradually released. The sustained release of CC nanoparticles is obvious, and the prolongation of release time can increase the effect of rhEGF on the skin surface, which is more conducive to the chance of interacting with skin cells and the growth of new tissue [34].

#### 3.3.2. Cell Compatibility of CC-NPs

MTT assay was processed on the L929 cell line to investigate the cell compatibility of CC-NPs. The MTT assay results were further confirmed by morphology observations of L929 cells incubated with CC3 nanoparticles at 24, 48, and 72 h, as is shown in Figure 5A. L929 cells retained normal morphology after incubation with CC3 nanoparticles for 3 days. The cell viability of each group was above 80% when the concentrations ranged from 250 to 2000 μg/mL (Figure 5B), which demonstrates that none of the CC-NPs are toxic, except for the maximum concentration (2 mg/mL) group [35]. There was no difference in cell viability between CC-NP groups, implying that the chemical modification did not disrupt the excellent cell compatibility of chitosan. Compared with many cationic polymer vectors that showed obvious cytotoxicity, CC-NPs are favorable for drug or gene delivery due to their extraordinary biocompatibility [36].

#### 3.3.3. Enhancement of Cell Proliferation and Migration by rhEGF:CC-NPs Treatment

MTT assay was used to calculate living cell numbers and reflect the bioactivity of rhEGF. Firstly, we used different concentrations (0, 40, 60, 80, 100 μg/L) of free rhEGF to verify the proliferation experiments of L929 cells (48 h). The absorbance values (OD_490 nm_) of different groups were 0.8187, 0.8204, 0.8606, 0.9964, and 0.9928. These results showed that there is a more obvious proliferation ability when the concentration was 80 μg/mL and there is almost no effect on the proliferation of cells when the concentration is 40 μg/mL. Therefore, we use CC3-NPs to carry these two different concentrations of rhEGF to explore the drug encapsulation efficiency and efficacy enhancement of nanocarriers. Then, L929 cells were incubated with 0 μg/L (control), 40/80 μg/L (rhEGF:CC-NPs), and 80 μg/L (free rhEGF) for 48 h. As can be seen from Figure 6, free rhEGF significantly stimulated cell proliferation at 80 μg/L; however, rhEGF:CC-NPs promoted proliferation more strongly than free rhEGF. This indicated that the encapsulation of rhEGF in CC nanocarriers enhanced its proliferation effect, and the proliferation effect of rhEGF:CC-NPs is dose-dependent.

The migration assay results verified the proliferation observations. As can be seen from Figure 7, L929 cells treated with free rhEGF or rhEGF:CC-NPs showed enhanced migration compared with the control group, and more and more obvious migration was observed when the time was extended. At 6 h, the extent of wound closure was (34.74 ± 2.18)% for free rhEGF, (38.62 ± 5.29)% for rhEGF:CC-NPs, and (11.27 ± 4.06)% for the serum-free medium control group. After 12 h, the advantage of rhEGF:CC-NPs over free rhEGF became more evident. For all the treated groups, including the free rhEGF group and rhEGF:CC-NPs groups, the differences with the control group are statistically significant. In vitro proliferation and migration assay indicates that cells treated with the free rhEGF and rhEGF:CC-NPs proliferated and migrated, closing the wound, and that rhEGF:CC-NPs achieved better wound closure than free rhEGF [37].

### 3.4. In Vivo Studies

#### 3.4.1. Change of Body Weight and Promotion of Wound Contraction

On day 14, no adverse reactions were observed in all groups of rats, and there was no significant difference in their body weight [38]. The skin regenerated rapidly to repair the wound (Table 2), and typical wound images were obtained from each treatment group at day 14 (Figure 8). Nearly half of the wound area was completely healed on the 3 days after rhEGF:CC-NPs treatment, and the contraction in the rhEGF:CC-NPs group was significantly different from the free rhEGF group ((80.89 ± 4.07)% and (73.53 ± 3.62)%, respectively) after 7 days. Therefore, the rate of wound repair in the rhEGF-treated group (free rhEGF and rhEGF:CC NPs) was higher than that in the control groups, while the wound area in those groups could not be reduced by the same percentage ((58.98 ± 3.02)% for the untreated control group and (61.67 ± 1.84)% for the empty CC-NPs group).

By day 11 post-wounding, the wound-closure rate in the rhEGF:CC-NPs group ((91.27 ± 1.63)%) was significantly higher than that of the other groups. Moreover, the wound was almost completely healed after rhEGF:CC-NPs treatment at day 14. Effective wound healing required a coordinated response of a variety of cells, including fibroblasts, keratinocytes, and vascular endothelial cells [39]. The results now showed that the topical application of rhEGF:CC-NPs could promote the proliferation and migration of fibroblasts to accelerate the healing of full-thickness skin wounds in rat (Figure 6 and Figure 7).

#### 3.4.2. Results of H&E Staining

On the 14th day, tissue sections stained by H&E showed that the wounds treated with rhEGF:CC NPs had recovered the same morphology as the healthy skin (Figure 9D,H). Compared with the empty CC-NPs and free rhEGF group, it showed a more complete cure [40]. In addition, as shown in Figure 9, the healing of the drug-loaded nanospheres was significantly better than the other groups with intact epidermal structure and rich fibroblasts in the granulation tissue. In particular, the number of normal epidermal cells increased, and the number of inflammatory cells decreased significantly. At the same time, it was observed that new thick-walled epidermis had formed, and collagen was present in the skin tissue. However, the epidermal structure of other groups was still incomplete (especially control and CC-NPs groups, as shown in Figure 9A,B). There was a lack of internal capillaries, fibroblasts, and granulation tissue. As previously reported by several authors [41], these results demonstrated that the rhEGF released from rhEGF:CC-NPs could promote wound healing by accelerating angiogenesis and inflammation and inducing epithelial formation and collagen accumulation. It should be noted that the differences between rhEGF:CC-NPs and free rhEGF indicated that the encapsulation process protected the growth factor against enzymatic degradation [7].

## 4. Conclusions

In this study, an efficient method for the hydrophobic synthesis of well-distributed spherical CC-NPs was developed. Conjugated linoleic acid was successfully loaded in CC-NPs after its cross-linking with carboxymethyl chitosan. The CC-NPs prepared in this way were uniform and stable in solution over a period of one month at room temperature and showed no signs of aggregation. The cross-linking showed no cytotoxicity and enhanced the loading of the selected unstable protein in NPs. Furthermore, the results demonstrated that the bioactivity of the rhEGF:CC-NPs was higher in L929 cell lines than that of free rhEGF. Due to the efficient encapsulation and control release, the epidermal full-thickness wound healing was promoted significantly. Finally, our study suggests the promising future clinical application of rhEGF-loaded CC-NPs for the treatment of chronic wounds. However, further studies are required to demonstrate the molecular mechanism to promote wound healing.

## Figures and Tables

**Figure 1 polymers-12-01612-f001:**
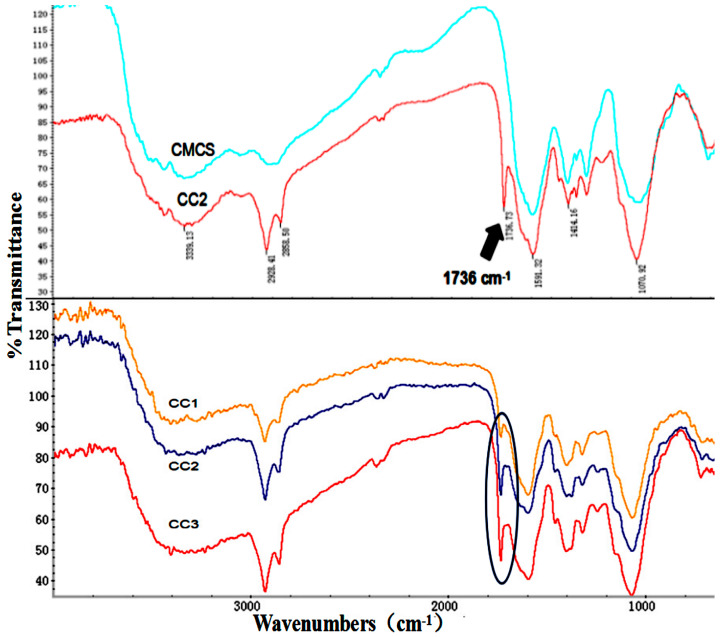
FTIR spectrum of carboxymethyl chitosan (CMCS) and CC1-3. CC: CLA - CMCS.

**Figure 2 polymers-12-01612-f002:**
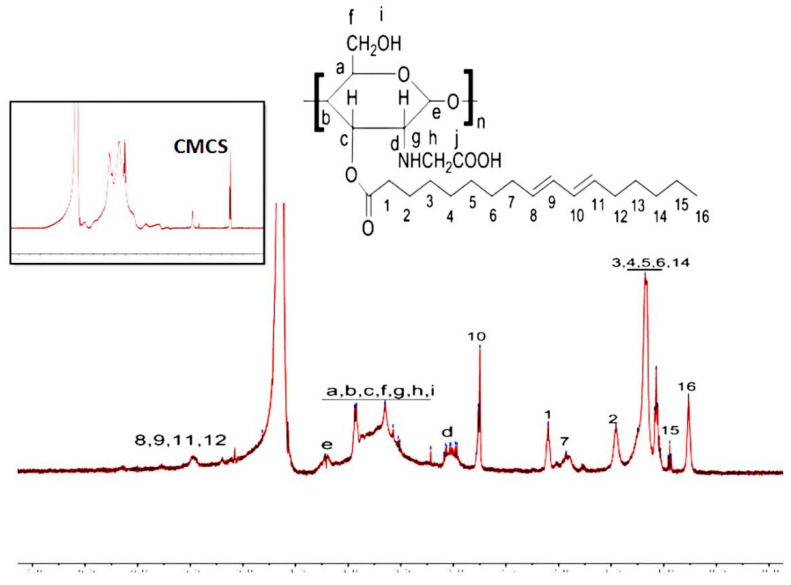
^1^H NMR spectra of CMCS and CC3.

**Figure 3 polymers-12-01612-f003:**
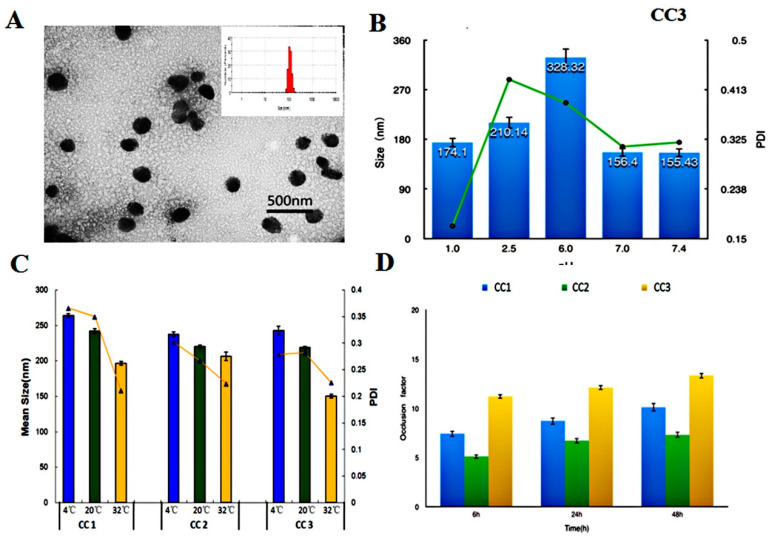
(**A**): TEM of self-aggregates based on CC3 conjugate; (**B**,**C**): The changes of size and PDI of CC-NPs (nanoparticles) at different pH (**B**) and temperature (**C**); (**D**): Occlusion factors of CC-NPs.

**Figure 4 polymers-12-01612-f004:**
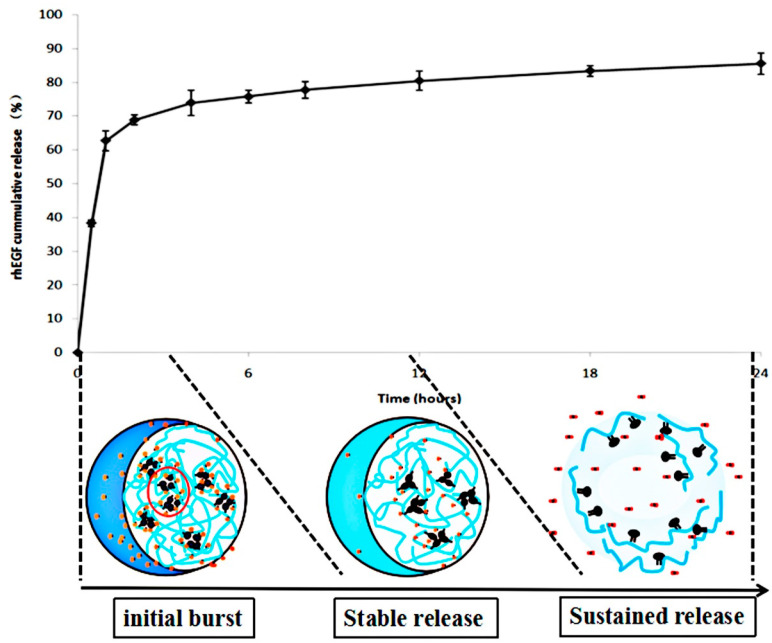
Cumulative in vitro release profile of recombinant human epidermal growth factor (rhEGF) from rhEGF:CC-NPs.

**Figure 5 polymers-12-01612-f005:**
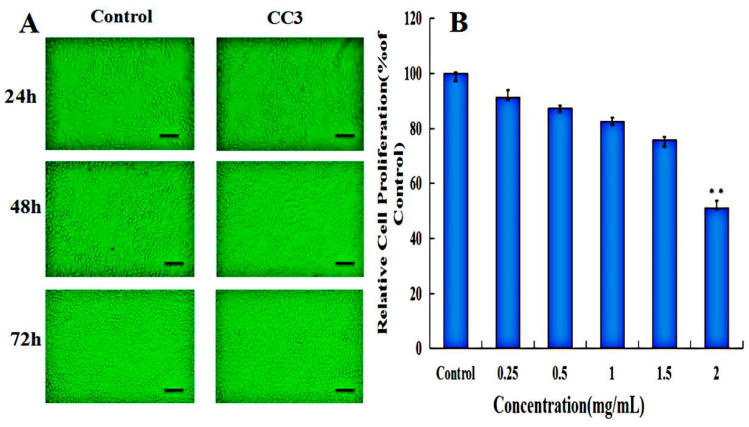
(**A**) Cell morphology of L929 cells treated without or with CC3 nanoparticles; (**B**) MTT assay of CC nanoparticles. Note: the scale bar represented 25 μm; ** *p* < 0.01 vs. control.

**Figure 6 polymers-12-01612-f006:**
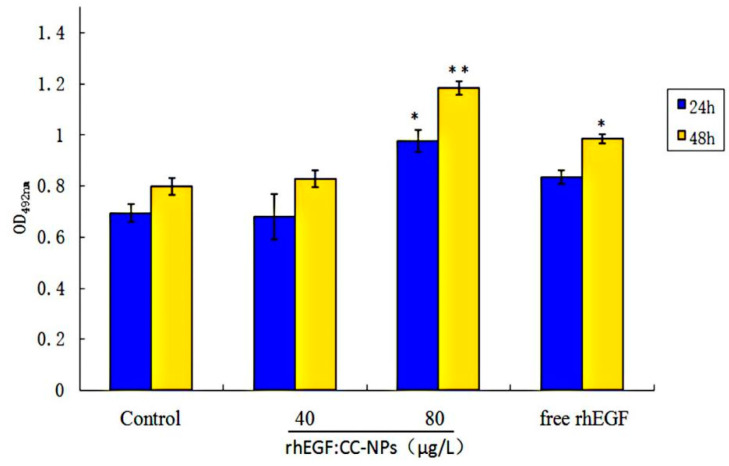
Proliferation of L929 cells when incubated with control, rhEGF:CC-NPs (40 μg/L), rhEGF:CC-NPs (80 μg/L), and free rhEGF (80 μg/L) for 24 and 48 h. Note: * *p* < 0.05 vs. control, ** *p* < 0.01 vs. control.

**Figure 7 polymers-12-01612-f007:**
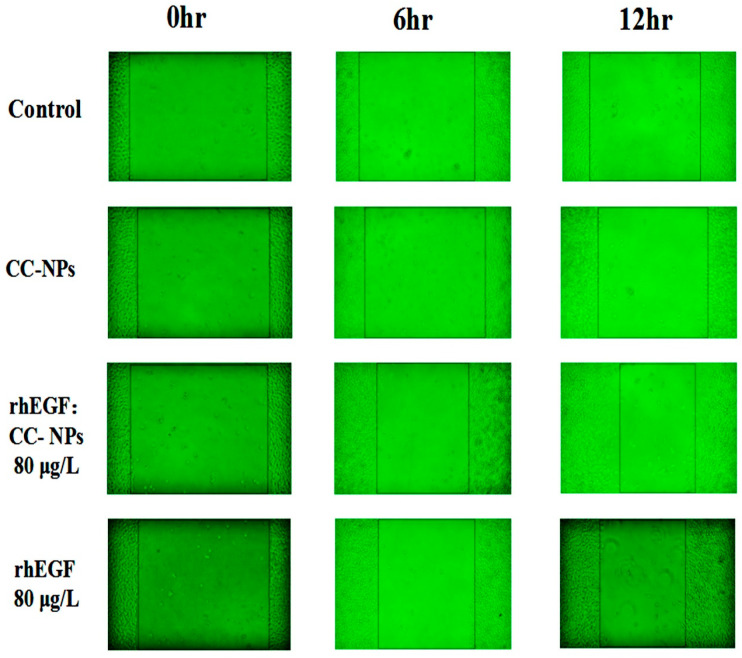
In vitro migration test.

**Figure 8 polymers-12-01612-f008:**
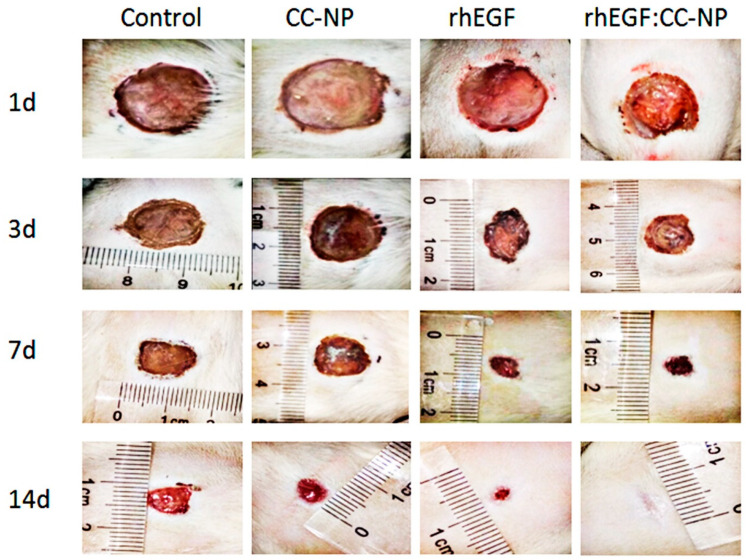
Wound images obtained from each treatment group.

**Figure 9 polymers-12-01612-f009:**
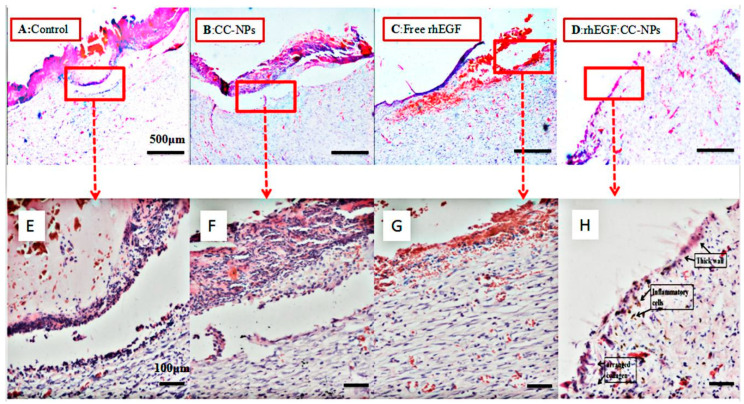
Hematoxylin and eosin (H&E) staining images of the wounds on the 14th day. (**A**) blank control group; (**B**) CC-NPs treatment group; (**C**) free rhEGF treatment group; (**D**) rhEGF:CC-NPs treatment group; (**E**–**H**) enlarged details of (**A**–**D**), respectively.

**Table 1 polymers-12-01612-t001:** General properties of CC conjugates and CC-NPs. DS: degree of substitution, LE: loading efficiency, PDI: polydispersity.

Formulation	DS (%)	Mean Size (nm)	PDI	Zeta Potential	Transmittance (%)	LE (%)	Hemolysis Rate (HR, %)
CC1	3.78	196.4 ± 5.49	0.272	−19.8 ± 3.07	72	72.16 ± 2.69	2.83
CC2	4.47	167.2 ± 11.38	0.239	−20.1 ± 1.05	55	78.68 ± 1.83	2.65
CC3	6.52	155.3 ± 4.62	0.202	−23.3 ± 0.37	93	82.43 ± 3.14	1.48

**Table 2 polymers-12-01612-t002:** Wound closure calculated as a percentage area of the original wound.

Groups	3rd d (%)	7th d (%)	11th d (%)	14th d (%)
Control	32.24 ± 1.96	58.98 ± 3.02	71.04 ± 4.62	80.86 ± 3.29
CC-NPs	29.47 ± 3.14	61.67 ± 1.84	73.49 ± 5.17	84.61 ± 4.06
rhEGF	39.10 ± 2.37	73.53 ± 3.62	83.90 ± 3.24	92.34 ± 2.17
rhEGF: CC-NPs	46.20 ± 4.08	80.89 ± 4.07	91.27 ± 1.63	98.61 ± 1.64

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
