# Peer review of "Enhancement of Skin Wound Healing by rhEGF-Loaded Carboxymethyl Chitosan Nanoparticles"

_polymers, 2020, doi:10.3390/polym12071612_

Round 1

Reviewer 1 Report

This research has highlighted on the generation method and therapeutic effects of carboxymethyl chitosan (CMCS)-based nanoparticles to delivery recombinant human epidermal growth factor (rhEGF) for the induction of the enhanced wound healing. To induce a long-term sustained release of rhEGF, CMCS was linked with conjugated linoleic acid (CLA) and developed nanoparticles (CC-NPs) were well characterized via various methods of analysis. Also, the bioactivity of CC-NPs showed improved effects both various in vitro experiments and in vivo wound healing studies. The experiments are well planned and the technical quality is not doubted. However, several issues need to be addressed in order for the manuscript to be published in Polymers. The reviewer offers following comments for improving the manuscript.

⦿ Major comments:

1. The values of zeta potential of generated CC-NP were measured with increased negative charge according to the increase of CLA feed ratio, while the size decreased. The surface charge of CC-NP with CCL3 was measured with the highest negative charge value, why is the best cellular uptake efficiency?

2. In Figure 5C, you mentioned that a large number of CC-NPs with CCL3 showed the best cellular uptake into L929 cells. What criteria do you refer to as being the best uptake? Compared to what, cellullar uptake is better? The control group for this experiment is CMCS particle, why didn't you proceed with the experiment, including the control group? It is necessary to demonstrate a comparative experiment with controls in cell viability or FITC cellular uptake experiment.

3. In cell proliferation assay, you used CC-NPs with two concentrations of rhEGF. The increased cell proliferation was not observed in the cells with treated with 40 ug rhEGF encapsulated CC-NPs. However, an increased effect showed when used 80 ug concentation of EGF. Why didn't you use a higher concentration of EGF? In other words, how did you determine the optimum concentration? Why did you use it up to 80 ug? If you do not have the results, please show the results of determining the optimal concentration

4. The purpose of this experiment is to overcome the shortcomings of EGF, which has a short half-life, and to maintain long-term therapeutic effects. In your in vivo study, therapeutic effects of free rhEGF was as long as your CC-NPs. How can you explain this ? Moreover, there is no explanation for concentration of CC-NP or rhEGF and method of administration for animal testing. How did you experiment?

5. In relation to Question 4, please arrow the points in Figure 9 about the blood vessel, arranged collagen, and the thick wall mentioned in this sentence (line 307 and 308 on page 12).

⦿ Minor comments:

1. Please write a coherent expression of words in Manuscript, like h or hour, ml or mL, or italic font of in vivo etc..

2. Please omit this term in subtitles of 2.4.2 or 2.4.3, as experimental subject 2.4 specifies in vitro.

3. Please add Reference in line 125 on page 3 or line 204 on page 7.

4. Please recheck Reference 19 (line 77 on page 2).

5. Please check the duplication of the content in from line 128 to 131 on page 3).

Author Response

Dear reviewer,

Thank you for your work to deal with our manuscript titled “Enhancement of skin wound healing by rhEGF loaded carboxymethyl chitosan nanoparticles” . 

Now, I provide the revised manuscript and cover letter to review comments of “Polymers-859825”. We thank you for the valuable comments and suggestions and we followed all of suggestions made. Any change to our manuscript has been highlighted in the revised paper. These comments helped us to improve our manuscript, and provided important guidance for future research.

Best regards !

Pei Zhang

Major comments:

Point 1: The values of zeta potential of generated CC-NP were measured with increased negative charge according to the increase of CLA feed ratio, while the size decreased. The surface charge of CC-NP with CCL3 was measured with the highest negative charge value, why is the best cellular uptake efficiency?

Response 1: Thank you very much for the comments and the first sentence is explained as follows:

Firstly,the values of zeta potential of generated CC-NP were measured with increased negative charge according to the increase of CLA feed ratio, while the size decreased. In this manuscript,the sizes of the majority of nanoparticles are 155–196 nm which is significantly affected by DS of CC conjugates. The particle size of the nanoparticles decreases with the increase of the CLA grafting rate, mainly due to the increase in the number of hydrophobic groups attached to the polysaccharide backbone and the hydrophobic force of the entire polymer is enhanced, which is conducive to the formation of a tighter core-shell structure. Therefore, the size of the CC3 nanoparticles is smaller than other.

Secondly, the increase of hydrophobic groups will inevitably change the Zeta potential on the surface of the nanoeparticles. The charge potential of the CC3 nanoparticles is -23.3±0.37 mV, which has the highest surface potential, and the nanoparticles are the most stable.

These above agree with the results obtained by Tan on FCC nanoparticles ( Tan Y L , Liu C G . Preparation and characterization of self-assemblied nanoparticles based on folic acid modified carboxymethyl chitosan[J]. J Mater, Mater Med, 2011, 22(5):1213-1220.) and Wei on HD nanoparticles (Wen-Hao, Wei, Xue-Meng,Dong, Chen-guang Liu. In Vitro Investigation of Self-Assembled Nanoparticles Based on Hyaluronic Acid-Deoxycholic Acid Conjugates for Controlled Release Doxorubicin: Effect of Degree of Substitution of Deoxycholic Acid[J]. International Journal of Molecular Sciences, 2015. ).

And, the answer to the second question (The surface charge of CC-NP with CCL3 was measured with the highest negative charge value, why is the best cellular uptake efficiency?) is as follows:

CC3 sample with the highest DS showed the strongest transmittance than others samples. The higher light transmittance indicated that the size of the nanoparticles was smaller and the degree of uniformity was higher, and there was no phenomenon of aggregation in the solution, which could be observed by TEM.

Finally, we are very sorry that we have neglected to describe details of transmittance methodology. We have added this part in the materials and methods section:

2.2. Synthesis of CLA-CMCS conjugates

CC-NPs solution (3 mL) were taken to determine the absorbance (OD) of the nanoparticle solution at a wavelength of 600 nm by spectrophotometry and use PBS buffer (pH 7.4) as a control to detect its light transmittance.

Point 2: In Figure 5C, you mentioned that a large number of CC-NPs with CCL3 showed the best cellular uptake into L929 cells. What criteria do you refer to as being the best uptake? Compared to what, cellullar uptake is better? The control group for this experiment is CMCS particle, why didn't you proceed with the experiment, including the control group? It is necessary to demonstrate a comparative experiment with controls in cell viability or FITC cellular uptake experiment.

Response 2: We are very sorry for providing an unclear experimental result and we made a decision to delete the following parts on the basis of carefully re-examining the original data, so as to maintain the integrity and accuracy of the entire work.

The content of the cell uptake experiment is deleted as follows:

Materials and Methods

In addition, to explore the cellular uptake of CC NPs was based on the previously reported method. Confocal laser scanning microscopy (CLSM) and fluorescent enzyme labeling (flx800b, bio TEK, USA) were needed for this uptake experiment.

Results and discussion

Nowadays, effective cell uptake can guarantee the successful application of nano drug delivery system in disease treatment. FITC labeled CC3-NPs (150 nm) were incubated with L929 cells for 4h before CLSM observation. As shown in Figure 5C, a large number of CC-NPs are uptake by L929 cells within 4h which was inferred to be an time, energy,size-dependent and saturable process38-40.

References

  1. Almalik A , Karimi S , Ouasti S , et al. Hyaluronic acid (HA) presentation as a tool to modulate and control the receptor-mediated uptake of HA-coated nanoparticles[J]. Biomaterials, 2013, 34(21):5369-5380.
  2. Ichikawa S, Shimokawa N, Takagi M, et al. Size-Dependent Uptake of Electrically Neutral Amphipathic Polymeric Nanoparticles by Cell-sized Liposomes and an Insight into Their Internalization Mechanism in Living Cells[J]. Chemical Communications, 2018, 54(36):4557-4569.
  3. Losi P , Briganti E , Errico C , et al. Fibrin-based scaffold incorporating VEGF- and bFGF-loaded nanoparticles stimulates wound healing in diabetic mice[J]. Acta Biomaterialia, 2013, 9(8):7814-7821.

In addition, cell cytotoxicity of CC-NPs was determined using an MTT assay. The culture medium was replaced with a new one containing blank CC-NPs, in which the concentration range of NPs were 50-1000 μg/mL. Cell survival was expressed as a percentage of the control (only test cells added). The cytotoxicity results are consistent with the results of our previous experiments (Zhang P, Guo H, Liu C G. Fabrication of Carboxylmethyl Chitosan Nanocarrier via Self-Assembly for Efficient Delivery of Phenylethyl Resorcinol in B16 Cells[J]. Polymers, 2020,12(2):408. / Pei Z, Yan Zhang, Liu C G. Polymeric nanoparticles based on carboxymethyl chitosan in combination with painless microneedle therapy systems for enhancing transdermal insulin delivery[J]. RCS Advances.2020, 10(41):24319. ).

And we are very sorry that we have neglected to provide the result of control group in cell viability. Finally, Figure 5 was changed as follows:

Figure 5. A: Cell morphology of L929 cells treated without or with CC3 nanoparticles; B:MTT assay of CC nanoparticles. Note:the scale bar represented 25μm; **P<0.01 vs control.

Point 3: In cell proliferation assay, you used CC-NPs with two concentrations of rhEGF. The increased cell proliferation was not observed in the cells with treated with 40 ug rhEGF encapsulated CC-NPs. However, an increased effect showed when used 80 ug concentation of EGF. Why didn't you use a higher concentration of EGF? In other words, how did you determine the optimum concentration? Why did you use it up to 80 ug? If you do not have the results, please show the results of determining the optimal concentration.

Response 3: We are sorry that we did not present the results of Cell proliferation of free rhEGF in the manuscript. We have added this part in the Results and discussion:

3.3.4. Enhancement of cell proliferation and migration by rhEGF:CC-NPs treatment

Firstly, we used different concentrations (0, 40, 60, 80, 100 μg/mL) of free rhEGF to verify the proliferation experiments of L929 cells (48 hours). The absorbance values (OD490nm) of different groups were 0.8187, 0.8204, 0.8606, 0.9964, 0.9928. These results showed that there is a more obvious proliferation ability when the concentration was 80 μg/mL and there is almost no effect on the proliferation of cells when the concentration is 40 μg/mL. Therefore, we use CC3-NPs to carry these two different concentrations of rhEGF to explore the drug encapsulation efficiency and efficacy enhancement of nanocarriers. 

In addition, the selection of rhEGF concentration also refers to related references (Gainza G , Pastor M , Aguirre, José Javier, et al. A novel strategy for the treatment of chronic wounds based on the topical administration of rhEGF-loaded lipid nanoparticles: In vitro bioactivity and in vivo effectiveness in healing-impaired db/db mice[J]. Journal of Controlled Release, 2014, 185:51-61. / 44.Choi J K , Jang J H , Jang W H , et al. The effect of epidermal growth factor (EGF) conjugated with low-molecular-weight protamine (LMWP) on wound healing of the skin[J]. Biomaterials, 2012, 33(33):8579-8590.).

Point 4: The purpose of this experiment is to overcome the shortcomings of EGF, which has a short half-life, and to maintain long-term therapeutic effects. In your in vivo study, therapeutic effects of free rhEGF was as long as your CC-NPs. How can you explain this ? Moreover, there is no explanation for concentration of CC-NP or rhEGF and method of administration for animal testing. How did you experiment?

Response 4: Thank you very much for the comments and the first question is explained as follows: 

rhEGF has the effect of accelerating epidermal reconstruction and growth (Kim H , Kong W H , Seong K Y , et al. Hyaluronate - Epidermal Growth Factor Conjugate for Skin Wound Healing and Regeneration[J]. Biomacromolecules, 2016,17(11):3694-3705.). However, rhEGF’s short half life requires a continuous exposure (at least 6–12 h) to enhance the mitogenic effect on epithelial cells (Hardwicke J , Schmaljohann D , Boyce D , et al. Epidermal growth factor therapy and wound healing--past, present and future perspectives.[J]. Surgeon, 2008, 6(3):172-177.). In this experiment, the nano-loaded group and the free drug group were administered once every 12 hours (twice per day) to mainly study the delivery and sustained-release functions of CC nanocarriers, thus the free group of drugs also maintained its efficacy.

And we are very sorry that we have neglected to describe the concentration of CC-NP or rhEGF and details of administration for animal testing. We have added this part in the materials and methods section:

2.5.2. Serial wound analysis

After the wound was induced, control group (PBS, pH=7.4), empty CC-NPs group, free rhEGF group (80 ug/mL) and rhEGF:CC-NPs group (80 ug/mL) were topically applied to the wound area twice per day for 14 days.

Point 5: In relation to Question 4, please arrow the points in Figure 9 about the blood vessel, arranged collagen, and the thick wall mentioned in this sentence (line 307 and 308 on page 12).

Response 5: According to the comments, we have corrected the result and legend.

In particular, the number of normal epidermal cells increased and the number of inflammatory cells decreased significantly. At the same time, it was observed that new thick-walled epidermis had formed and collagen was present in the skin tissue.

Figure 9. H&E staining images of the wounds on the 14th day.

Minor comments:

Point 1: Please write a coherent expression of words in Manuscript, like h or hour, ml or mL, or italic font of in vivo etc..

Response 1: We are very sorry that we ignored this kind of detailed error, and we carefully checked the full manuscript.

L929 cells were cultured in DMEM supplemented with 10% heated-inactivated fetal bovine serum (FBS), 100 U/mL penicillin, 100 ug/mL streptomycin and 2 mM glutamine.

L929 cells at 1×105 cells/mL were seeded onto 96-well plates in DMEM and cultured at 37 °C for 24 hours to form a confluent monolayer.

Furthermore, cell morphology was recorded during 24 hours by microscope (Biostation IMQ, Nikon) in order to observe cell migration and proliferation.

It exhibited an initial burst of 68.84 ± 1.53% in 2 h, and an accumulative release of 85.50 ± 3.08% after 24 hours.

After co-incubation for 4 hours, L929 cells were washed with D-Hanks and observed with Olympus fluorescence microscope.

For example, EGF is an important kind of GF which can promote proliferation of keratinocytes in vivo and accelerate wound healing9.

Firstly, CMCS was dissolved in aqueous solution at a final concentration of 1% (w / v).

The in vitro occlusion test was adapted from vringer's protocol and repeated three times (n = 3).

The in vitro hemolysis assay verified the good biocompatibility of CC-NPs, with CC3 the most biocompatible sample. 

Point 2: Please omit this term in subtitles of 2.4.2 or 2.4.3, as experimental subject 2.4 specifies in vitro.

Response 2: According to the comments, the subtitles of 2.4.2 or 2.4.3 have been replaced:

2.4.2. Cell compatibility assay 

2.4.3. Migration assay

Point 3:  Please add Reference in line 125 on page 3 or line 204 on page 7.

Response 3: According to the comments,we have added the reference in line 125 ï¼ˆCell cytotoxicity of CC-NPs was determined using an MTT assay12.)and line 204 on page 7(These results above agree with the results obtained by E.B. Souto on lipid nanoparticles21. ).

Point 4:  Please recheck Reference 19 (line 77 on page 2).

Response 4: According to the comments, the original reference 19 has been replaced:

  1. Zhang P, Guo H, Liu CG. Fabrication of Carboxylmethyl Chitosan Nanocarrier via Self-Assembly for Efficient Delivery of Phenylethyl Resorcinol in B16 Cells[J]. Polymers, 2020,12(2):408.

Point 5: Please check the duplication of the content in from line 128 to 131 on page 3).

Response 5: We are very sorry that we ignored this kind of detailed error, and we delete this sentence:L929 cells were seeded on 96-well plates at a concentration of 1 × 105 cells / mL. In addition, we carefully checked the full manuscript.

Finally, thank you for these important suggestions. The authors are grateful to the referee for pointing out their error. Special thanks to you for your good comments!

Reviewer 2 Report

This manuscript has already clarified the application of CLA - CMCS polymer as a drug delivery system of rhEGFs. Some suggestions are provided for improvement:

  1. The materials and experimental methods should be illustrated more clearly for the reproducibility. For example, the D-solvent utilized in polymer characterization should be included. The TEM staining reagents should also be mentioned in this manuscript. Besides, the 
  2. In section 2.4.1, the authors utilized a spectrometer for rhEGF determination. However, the drug loading contents were determined at the wavelength of 243 nm, while the drug releasing profiles were measured at the wavelength of 275 nm. 
  3. Line 14: "Conjugated" should be "conjugated"

Author Response

Dear reviewer,

Thank you for your work to deal with our manuscript titled “Enhancement of skin wound healing by rhEGF loaded carboxymethyl chitosan nanoparticles” . 

Now, I provide the revised manuscript and cover letter to review comments of “Polymers-859825”. We thank you for the valuable comments and suggestions and we followed all of suggestions made. Any change to our manuscript has been highlighted in the revised paper. These comments helped us to improve our manuscript, and provided important guidance for future research.

Best regards !

Pei Zhang

Point 1: The materials and experimental methods should be illustrated more clearly for the reproducibility. For example, the D-solvent utilized in polymer characterization should be included.

Response:We thank to the reviewer for the valuable comment and we followed all of suggestions made. According to the comment, we have added that the D-solvent utilized in polymer characterization should be included:

  1. Materials and Methods

2.2. Synthesis of CLA-CMCS conjugates

The CC conjugates were characterized by FTIR and 1H NMR. The IR spectra of lyophilized CC conjugates were measured on a Nexus 470 FTIR spectrometer (Nicolet, USA) at 25 °C. Powder samples were blended with high quality and dry KBr, and then pressed into disks for measurement. The 1H NMR spectrum of the conjugates was generated by a 500-MHz NMR (UNIYTINOVA-500 NMR, VARIAN) at 25 °C. CC sample was dissolved in solution of NaOD (analytical reagent, Sigma), yielding a concentration of 10 mg/mL. The DS of CC conjugates were determined by elemental analysis (C, N) using a CE-440 elemental analysis instrument. 

2.3. Preparation and characterization of CC-NPs

The CC conjugates were repeatedly dissolved in distilled water and stirred for 24 hours. 

The morphology of CC-NPs was observed by transmission electron microscopy (TEM, H-600a, Hitachi, Japan). The sample was diluted with distilled water and placed on a copper mesh, which were air dried and negatively stained with 1% (w/v) phosphotungstic acid before observation. The CC-NPs were dispersed in 0.2 M phosphate buffered solution (PBS, pH 7.4) and sonicated to determine the particle size, polydispersity (PDI), and Zeta potential using a Malvern Zetasizer (3000HSA, Malvern, UK) at 25°C.

Point 2: The TEM staining reagents should also be mentioned in this manuscript.

Response:We are very sorry that we have neglected to describe staining reagents of TEM. We have added this part in the materials and methods section:

  1. Materials and Methods

2.3. Preparation and characterization of CC-NPs

The morphology of CC-NPs was observed by transmission electron microscopy (TEM, H-600a, Hitachi, Japan). The sample was diluted with distilled water and placed on a copper mesh, which were air dried and negatively stained with 1% (w/v) phosphotungstic acid before observation. The mean size, polydispersity index (PDI), and Zeta potential of CC-NPs were determined by zetasizer (Malvern 3000HSA, UK) at 25°C.

Point 3: In section 2.4.1, the authors utilized a spectrometer for rhEGF determination. However, the drug loading contents were determined at the wavelength of 243 nm, while the drug releasing profiles were measured at the wavelength of 275 nm.

Response:We are very sorry for that we provide the wrong wavelength in section 2.4.1. Now, we have corrected the wavelength of 275 nm as follows:

After centrifugation, the content of rhEGF was determined at 275 nm by uvikon 940 spectrophotometer (Kontron instruments, eching, Germany).

The samples were collected at different time points in a day (250 μL each time) and finally detected at 275 nm using the spectrophotometer.

Point 4:   "Conjugated" should be "conjugated"

Response:We are very sorry that we ignore such a obvious error in the sentence. And we have carefully checked and corrected the entire wrong words.

The amphiphilic polymer was composed of conjugated linoleic acid (CLA) and carboxymethyl chitosan (CMCS), which were further characterized by FT-IR and 1H NMR. 

Finally, thank you for these important suggestions. The authors are grateful to the referee for pointing out their error. Special thanks to you for your good comments!

Round 2

Reviewer 1 Report

Authors revised to reflect some of my comments.

This paper can be accepted for publication as it is now.